# Sleep Pattern Analysis in Unconstrained and Unconscious State

**DOI:** 10.3390/s22239296

**Published:** 2022-11-29

**Authors:** Won-Ho Jun, Hyung-Ju Kim, Youn-Sik Hong

**Affiliations:** Department of Computer Science and Engineering, Incheon National University, Incheon 22012, Republic of Korea

**Keywords:** sleep pattern, sleep posture, NREM, REM, unconscious, unrestraint

## Abstract

Sleep accounts for one-third of an individual’s life and is a measure of health. Both sleep time and quality are essential, and a person requires sound sleep to stay healthy. Generally, sleep patterns are influenced by genetic factors and differ among people. Therefore, analyzing whether individual sleep patterns guarantee sufficient sleep is necessary. Here, we aimed to acquire information regarding the sleep status of individuals in an unconstrained and unconscious state to consequently classify the sleep state. Accordingly, we collected data associated with the sleep status of individuals, such as frequency of tosses and turns, snoring, and body temperature, as well as environmental data, such as room temperature, humidity, illuminance, carbon dioxide concentration, and ambient noise. The sleep state was classified into two stages: nonrapid eye movement and rapid eye movement sleep, rather than the general four stages. Furthermore, to verify the validity of the sleep state classifications, we compared them with heart rate.

## 1. Introduction

Philips announced the findings of its sixth annual sleep survey in a report titled “Seeking Solutions: How COVID-19 Changed Sleep Around the World” [1]. Nearly a year after the onset of the coronavirus disease 2019 pandemic, Philips surveyed 13,000 adults in 13 countries to record their attitudes, perceptions, and behaviors regarding sleep. This year’s survey revealed that the global weekday and weekend sleep durations were 6.9 and 7.8 h, respectively. Specifically, Koreans slept 6.7 h on weekdays and 7.4 h on weekends. In terms of major sleep problems, 70% of global respondents cited waking up in the middle of the night (43%), an inability to fall asleep (34%), and experiencing difficulties in maintaining sleep (27%). Only 4 of 10 Koreans were satisfied with their sleep, with the response rate of obtaining adequate sleep being the lowest in Korea among the 13 surveyed countries [1].

In general, 6–8 h of sleep a day is necessary to maintain a healthy daily life. A lack of sleep affects concentration and exercise ability, owing to accumulated fatigue. In addition, studies have shown that a decrease in sleep time can increase the risk of obesity. However, sleeping for longer than the appropriate duration does not produce any further beneficial effects either. Conversely, studies have reported that excessive amounts of sleep can be harmful for one’s health. Nonetheless, adequate sleep time may vary with age and does not apply equally to everyone [2].

To obtain a good night’s sleep, a basic sleep time should be guaranteed. Individuals can control their sleep duration according to their will. However, sleep quality is more important than sleep time. Unfortunately, the depth of sleep, which determines the quality of sleep, cannot be controlled by the individual’s will. Although sleep quality is difficult to control, it can help prevent poor sleep quality by eliminating factors that interfere with a good night’s sleep [3]. Additionally, analyzing whether sleep patterns ensure sufficient sleep is necessary.

During sleep, our body cycles through four stages, comprising both rapid eye movement (REM) and nonrapid eye movement (NREM) sleep [4] (Figure 1). Stage 1 constitutes the REM sleep state, indicating a state of light sleep, whereas stage 4 is the NREM sleep state, indicating a state of deep sleep. During normal sleep, the stages follow a structured sequence, starting with wake, then light sleep with stages 1 and 2, followed by deep sleep (slow-wave sleep) with stages 3 and 4, and finally REM sleep. On average, the body cycles through these stages four to six times, ~90 min for each stage. As the night progresses, fewer NREM stages occur, and the duration of REM sleep increases [5]. A normal night comprises six sleep cycles, where the proportion of deep sleep decreases from the beginning to the end of the night, and the proportion of REM sleep increases concurrently [6].

Herein, we proposed a prototype system that can be used to determine whether an individual sleeps well by analyzing their sleep patterns. Accordingly, we attempted to determine sleep-related information to characterize a good night’s sleep. The proposed sleep monitoring system determines only whether an individual is in an REM or NREM sleep state (i.e., one of the three stages of NREM) and does not accurately distinguish the four sleep stages.

Data associated with sleep status, such as the frequency of tossing and turning, snoring, and body temperature, were directly related to the concerned individual. Conversely, environmental data, including room temperature, humidity, light, carbon dioxide (${\mathrm{CO}}_{2}$) concentration, and ambient noise, which affect a good night’s sleep, were not directly linked to individuals. A system involving a smart pillow with built-in pressure sensors was employed to acquire these sleep-related data, including the degree of tossing and turning, in the unconstrained and unconscious state. Furthermore, a contactless temperature sensor attached to the ceiling was used to measure body temperature.

The primary advantage of the proposed monitoring system is that it can monitor the sleep state of individuals in daily life without any restraint. However, it cannot accurately distinguish sleep states, as it does not acquire any biometric information.

Thus, this study aimed to collect unbiased and customized data over a long period rather than for an extremely short period in multiple participants. One of the authors was a participant, and experimental data were collected over 2 years. The experiments were conducted in our laboratory to maintain the experimental environments. This paper is an extended version of a conference paper [7].

This paper is organized as follows: Section 2 describes related works that distinguish between sleep states. Section 3 explains the overall configuration of the proposed sleep pattern monitoring system (SPMS). Section 4 describes the set of contactless devices used to acquire sleep state information and presents the experimental results. Section 5 provides concluding remarks.

## 2. Related Works

As mentioned in the previous section, sleep quality is important for an individual’s health. Polysomnography (PSG) is the most common method of assessing sleep quality. However, it is intrusive and expensive, requiring extensive expertise for public use. Therefore, several previous studies aimed to develop less expensive and nonintrusive methods to assess sleep quality.

Studies related to sleep analysis can be divided into sleep movement analysis [8,9,10], sleep disorder (or sleep disturbance) analysis [8,11,12], and sleep stage analysis [4,13,14,15]. Accordingly, the main analysis targets include breathing [11,16], body movement [8,9], heart rate [13,16,17,18], and respiration rate [16]. However, as important vital signs, including heart rate, electrocardiogram (ECG), and PSG, cannot be measured noninvasively, they are excluded from recent studies.

The primary equipment used for sleep monitoring is *imaging technology*, such as Kinect [3], an RGB camera [12], a thermal imaging camera [9,11], a pillow with built-in force sensitive resistor (FSR) sensors [19,20], and sensors attached to the arm or stomach (wrist sensor [10,15,21], flexible belt [17]). Several studies have also employed transmission methods (wireless fidelity [Wi-Fi] [19] and ultrawide band [UWB] [18]).

Widasari et al. [13] focused on obstructive sleep apnea (OSA), a potentially serious sleep disorder. It causes repeated cessations of breathing during sleep. The authors used only ECG signals, which are easy to conduct and record. Heart rate variability (HRV) spectrum analysis was applied to feature extraction, and a decision tree-based support vector machine classifier was used to measure the four parameters of sleep quality: sleep onset latency, total sleep time, sleep efficiency, and delta-sleep efficiency based on 30-s segments of ECG signals. Sleep quality was estimated using the automatic sleep stage and then was compared with PSG data.

Chung et al. [15] presented an approach to classify sleep stages via a low cost and noncontact multimodal sensor fusion, which extracted sleep-related vital signals from radar signals and a sound-based context-awareness technique. Furthermore, they incorporated medical/statistical knowledge to determine personal-adjusted thresholds and device postprocessing and compared sleep stage classification performance between a single sensor and sensor fusion algorithms.

Li et al. [21] developed the smart pillow to provide a relatively easy method of observing sleep conditions, including temperature and humidity, by strategically implanting the respective sensors inside the pillow. They extracted sleep patterns via statistical analysis, and the body temperature was inferred via fuzzy logic if the head-on position was stable for >15 min.

Lee et al. [22] proposed a sleep monitoring system that can detect sleep movement and posture using a Microsoft Kinect v2 sensor (Redmond, WA, USA) without any wearable device. However, in the proposed system, the depth sensor does not work if a blanket is covering the human body. Liu et al. [19] proposed Wi-Sleep, a sleep monitoring system based on Wi-Fi signals, which continuously collects fine-grained wireless channel state information (CSI) around a person. From the CSI, it extracts rhythmic patterns associated with respiration and abrupt changes due to body movements.

A study [14] proposed a classification algorithm for the sleep stages (four stages of “wake”, “light sleep”, “deep sleep”, and “rapid eye movement”) based only on wrist movements acquired using an accelerometer.

Lin et al. [8] developed a noncontact and cost-effective sleep monitoring system, SleepSense, for the continuous monitoring of sleep status, including on-bed movement, bed exit, and breathing section. It constitutes three parts: a radar-based sensor, radar demodulation module, and sleep status-recognition framework. It extracts several time- and frequency-domain features for the sleep-recognition framework. Huang et al. [11] proposed a classification of nasal and mouth breathing using the thermography of the participant. The measurement used the relative temperature variations of different facial regions to classify mouth or nasal breathing. This measurement is particularly relevant to the health and well-being of individuals, as it can be used to detect early signs of sleep disorders or indicate sleep quality.

Jakkaew et al. [9] presented the noncontact respiration and body movement monitoring system. Automatic region of interest extraction via temperature and breathing motion detection is based on integrated image processing to obtain respiration signals. As thermal imaging cameras have various viewing angles, they are easy to install in bedrooms. A signal processing technique is used to estimate respiration and body movement from a sequence of the thermal video. Chen [12] developed a noninvasive sleep monitoring system to distinguish sleep disturbances. The prototype system contains an infrared depth sensor, RGB camera, and four-microphone array to detect three events, namely motion, lighting, and sound events.

Li et al. [16] proposed a method of using simple signals, such as heart and respiration rates, and integrated prior knowledge of experts to simplify the feature extraction process, thereby enabling the subsequent classifier to easily distinguish between the waking and sleep states. Siyang et al. [19] developed an Internet of Things (IoT) solution to monitor sleep based on a data pillow system. They installed FSRs under the pillow to collect breathing data, reporting that the IoT data pillow can detect breathing signal differences among normal respiration, hypopnea, and apnea.

Bao et al. [10] proposed a noncontact human sleep monitoring method that characterizes sleep stages via two aspects of body motion and respiration and compares them with the data acquired by traditional wristband products. Veiga et al. [20] proposed an IoT-based sleep quality monitoring pillow that tracks temperature, humidity, luminosity, sound, and vibration. He et al. [17] presented a flexible sleep monitoring belt with a microelectromechanical system triaxial accelerometer and pressure sensor to detect vital signs, snoring events, and sleep stages. They tried to detect heart and respiration rates, to recognize snoring, and to classify sleep stages. The test results measured by PSG were used as the gold standards for comparison.

Im et al. [18] proposed a noncontact sleep monitoring system using UWB and a photoplethysmogram (PPG). The proposed system comprised a UWB radar, environmental sensor board, and PPG sensor. The UWB radar measures the sleep-breathing and heart rates and movements of the user. The PPG sensor measures the heart rate and movements. A relay board equipped with an environmental sensor measures the ambient temperature, humidity, and illuminance data. However, the UWB radar has limitations, i.e., it is immediately vulnerable to posture and movements. In case of incorrect posture, inaccurate biometric data are measured, and in case of movement, no biometric data are measured.

Renevey et al. [4] presented a method to classify sleep phases using a wrist-worn device. Raw signals from PPG and a three-dimensional accelerometer were used. The PPG was used to measure the HRV, whereas the accelerometer was used to assess body movements. In iSleePost [23], the care recipient wears one accelerometer on his/her chest for automatic label collection and another on the wrist to collect movement data, and they attempted to recognize sleep postures using easy-to-wear wrist sensors.

## 3. Design of the SPMS

To minimize the effect of experimental environments on the daily sleep pattern of the participant, a system was built to monitor their sleep patterns in unconscious and unconstrained states. Figure 2 shows the sleep-related information that needs to be collected for sleep pattern analysis. Seven types of sensors were used to acquire the information (Figure 2). A noncontact temperature sensor, infrared camera, and light sensor were installed on the ceiling. A ${\mathrm{CO}}_{2}$ sensor, fine dust sensor, and sound detection sensor with a microphone were installed on the side of the bed. Additionally, the participant laid on the bed on a smart pillow with built-in pressure sensors. Each sensor sampled raw data every second.

A smart pillow with built-in pressure sensors was used to discriminate the sleeping postures of the participant and to determine the duration of each posture. In addition, this device was used to determine the quality of sleep by analyzing the number of tosses and turns during sleep. To verify the accuracy of sleeping posture discrimination using the smart pillow, the relevant sleeping posture discrimination data were compared with the sleeping posture images captured by the infrared camera.

When sounds such as snoring, teeth grinding, and ambient noise exceeding the specific dB (~30 dB) were generated near the bed, the sound detection sensor identified them and recorded them for 15 s using the sound-recording module. Habitual snoring is a prevalent condition that is not only a marker of OSA but can also lead to vascular risk [21]. Snoring may be a sign of a serious underlying sleep-related breathing disorder. Thus, when a factor such as snoring interferes with sleep, the recorded sound not only depicts a change in the sleep status but also enables the evaluation of the participant’s health status.

Conversely, the ${\mathrm{CO}}_{2}$ concentration generated during respiration may directly affect the sleep state. The ${\mathrm{CO}}_{2}$ sensor measured changes in the ${\mathrm{CO}}_{2}$ concentration during sleep to investigate whether the number of tosses and turns or snoring of the participant were related to changes in its concentration.

Sleeping posture, number of tosses and turns, body temperature, ${\mathrm{CO}}_{2}$ concentration, and recorded sounds constituted the sleep-related information that was obtained directly from the participant. Conversely, as environmental data, including room temperature, humidity, and concentration of fine dust, indirectly affect sleep quality, these data were used as auxiliary factors during sleep pattern analysis. In the near future, they can be fully utilized for air conditioning control.

A wrist-attached heart rate sensor was used to compare the sleep patterns of the participants using the sleep-related information obtained during the unconstrained and unconscious state along with the biometric information of the participant. Finally, for future applications, an additional device that plays music to induce psychological stability and deep sleep was deployed in the system. Figure 3 shows the overall architecture of the SPMS using nonwearable sensors. Table 1 summarizes the functions and detailed specifications of the sensors used in the system.

### 3.1. Sensors for Acquiring Sleep-Related Data

#### 3.1.1. Smart Pillow

To determine the sleeping posture, a smart pillow incorporated with eight pressure sensors (FSR-406) in a straight line was used (Figure 4). An FSR sensor detects pressure, weight, or touch applied to the sensor unit [24]. It has a rectangular shape (39.6 mm × 39.6 mm) and can measure a relatively wide pressure range. Euryon is the lateral-most point placed on the side of the head. The head breadth is the horizontal distance between the right euryon and the left euryon of the head. In Korea, it measures 168 mm for men and 159 mm for women [25]. The horizontal length of the FSR sensor array is 349.52 mm; thus, the smart pillow can measure various pressures generated from the head when changing sleeping postures. Furthermore, it can discriminate among three distinct sleeping postures (supine posture, lying to the left, and lying to the right) based on the pressure distribution (Figure 4b).

Information that can be measured using the smart pillow includes sleep time, sleeping posture, and the number of tosses and turns. Sleeping posture was determined by applying a machine learning model created using a support vector machine (SVM) to the pressure dataset. The dataset comprised over one million pressure samples collected using the smart pillow for a period of ~5 months (from September 2020 to January 2021).

#### 3.1.2. Infrared Camera Module

An infrared camera module was installed to verify the discrimination correctness and detection accuracy of the sleeping posture and number of tosses and turns during sleep, respectively, of the participant. This module was attached to the ceiling to capture sleeping posture images (Figure 5a), wherein it can capture images within a designated space without light (Figure 5b). While the participant was sleeping, it captured and stored images every second, which was the same as the sampling interval of the pressures using the smart pillow.

#### 3.1.3. Contactless Body Temperature Sensor

To measure the body temperature of the participant, the contactless temperature sensor (DTS-L300-V2) was installed on the ceiling (Figure 6). It can measure the body temperature of a person up to 1 m away, and body temperature, room temperature, and humidity can be measured simultaneously. During our experiments, the sensor was affected by an indoor temperature of up to ±2.0 °C according to the changes in season (summer and winter). The variation of ±2.0 °C at maximum should be calibrated in the monitoring system or controller (Arduino board).

#### 3.1.4. Sound Sensor Detection and Sound-Recording Module

The hardware circuit was configured (Figure 7) to detect and record the sounds (snoring, bruising, or ambient noise) generated by the participant during sleep. The sound pressure levels emanating from a quiet room and normal conversation at 4 feet away are 40 and 60 dB, respectively [26]. According to [27], the noise criterion for a bedroom is 30 dB. However, since the sound of snoring varies from person to person, this value needs to be adjusted according to the participant. Our experimental results showed that, if the sound level detected by the sound sensor exceeded the threshold (30 dB), it can be considered snoring. When the sensor detects a sound beyond the threshold, the sound card connected to the Raspberry-Pi board is activated and records it via the microphone. Sounds exceeding the threshold were recorded for 15 s.

The sound detection sensor detects the signal from the microphone, amplifies it, and converts it into a corresponding analog signal. With an increase in signal amplification, the sensitivity to detect it increases. Such sensitivity can be adjusted by controlling the variable resistance of the sound sensor.

### 3.2. Sensors for Acquiring Environmental Data

Body temperature changes are closely related to sleep initiation and maintenance. Body temperature is influenced by various factors, such as air temperature, room temperature, clothing, human activities, and meditation [28]. Extremely high and low environmental temperatures affect the sleep quantity and quality of healthy individuals [29]. Therefore, temperature is crucial for sleep regulation. Changes in environmental temperature not only affect sleep duration but also slow-wave and REM sleep states [28].

#### 3.2.1. Contactless Room Temperature and Humidity Sensors

Herein, only room temperature and humidity were considered limitations of the research scope. To measure them while the participant was asleep, contactless temperature and humidity sensors (DHT22) were installed (Figure 6).

#### 3.2.2. Light Sensor

When the human eye detects light, the brain secretes cortisol. It awakens the human body and affects the sleep state. Conversely, when the surrounding environment becomes dark, melatonin is secreted, which induces sleep. Light affects melatonin secretion in two ways: day–night light cycles modify the rhythm of its secretion and brief pulses of light of sufficient intensity and duration abruptly suppress its production [30]. Thus, for proper hormone function and a comfortable sleep state, the immediate environment must be kept dark when sleeping.

A photosensitive sensor measures ambient brightness. In this device, electrons alter the sensor conductivity when they receive light. As its conductivity does not increase linearly in proportion to brightness, it is suitable for determining only light or dark levels rather than obtaining an accurate value in Lux.

#### 3.2.3. CO_2_ Sensor

The most common indicator of indoor air quality is the concentration of ${\mathrm{CO}}_{2}$, as its presence in an indoor environment is strictly related to respiration and human metabolism [31]. Li et al. [32] showed that a high ${\mathrm{CO}}_{2}$ concentration stimulates the central nervous system to sustain dreams and exciting conditions by engendering the REM sleep split phenomenon. In other words, when it is high, the frequency of REM sleep is much higher compared with other times. Thus, the sleep environment can be characterized by ${\mathrm{CO}}_{2}$ concentration.

When the ${\mathrm{CO}}_{2}$ concentration in the room is 700–1000 ppm, an individual experiences an unpleasant feeling; at 1000–2000 ppm, changes in physical condition occur, such as fatigue and sleepiness; at >2000 ppm, headaches and shoulder stiffness develop; and at >3000 ppm, harmful effects may occur, such as dizziness [33]. The United States, Japan, China, and Korea recommend the indoor environment ${\mathrm{CO}}_{2}$ concentration to be <1000 ppm. The World Health Organization [34] suggests a maximum ${\mathrm{CO}}_{2}$ concentration of 1500 ppm but recommends maintaining the indoor ${\mathrm{CO}}_{2}$ concentration at <1000 ppm.

A device for measuring ${\mathrm{CO}}_{2}$ concentration (Figure 8) was installed on the left side of the bed. It was classified into four levels according to the concentration: good, <400 ppm; normal, <700 ppm; bad, <1000 ppm; and very bad, >1000 ppm. If it was above the bad level, the red LED turned on, as shown in Figure 8.

### 3.3. Wearable Sensor for Heart Rate Measurement

Heart rate is regulated by the autonomous nervous system, and sympathetic tone is strongly influenced by sleep stages [35]. An investigation of HRV provides insights into the mechanisms of sleep regulation. Penzel [35] showed that, when the brain is very active, as during the REM stage, heart rate has long-time correlations, such as in the wake phase. Conversely, during deep sleep, the correlations of the heart rate disappear after a small number of beats. Casal [6] proposed an automatic system to determine if a patient is awake or asleep on the basis of heart rate signals obtained via pulse oximetry.

Based on the results of previous studies, we distinguished REM and NREM sleep stages using heart rate, which was the only biometric information we could obtain. NREM sleep is characterized by slow heart rate and less fluctuations, whereas during REM sleep, heart rate increases. These classifications and results that were predicted using values measured during in the unconstrained and unconscious state were compared. The band-type heart rate-measuring device (Figure 9) determined the heart rate by measuring the movement of the blood based on optical technology.

## 4. Sleep Pattern Analysis in the Unconstrained and Unconscious State

We attempted to determine the sleep state of the participant while maintaining the unconstrained and unconscious state in the usual living space. Ambient environmental data, including light, room temperature, and ${\mathrm{CO}}_{2\;}$ concentration, were believed to exert a significant effect on the sleep state; however, experimental results that artificially adjusted these values were excluded from the sleep state analysis because such artificial adjusting in environmental data is not suitable for our research purpose of establishing a non-restraining and unconscious environment.

The proposed monitoring system determined the sleeping state by comprehensively analyzing the body movement (smart pillow), condition (body temperature), and body (snoring, teeth grinding, etc.) of the participant. For time t, the sleep state $Sleep_{state}\left(t\right)$ was correlated with the above data, as shown in Equation (1).
(1)$$Sleep_{state}\left(t\right)=f\left(g_{toss-and-turn}\left(t\right),\;g_{snoring}\left(t\right),\;g_{temperature}\left(t\right)\right)$$

In Equation (1), $g_{toss-and-turn}\left(\;t\right)\;$ denotes the number of tosses and turns, $g_{snoring}\left(\;t\right)\;$ denotes the number of times snoring was detected during sleep, and $g_{temperature}\left(\;t\right)$ is a function related to the body and room temperature.
(2)$$g_{toss-and-turn}\left(\;t\right)\propto (1/\left(T_{i}^{posture}\left(t\right)\right)\;for\;i\in N_{toss-and-turn}$$

In Equation (2), $N_{toss-and-turn}$ denotes the number of times the sleeping posture was changed. The sleeping postures were divided into three patterns: supine, lying to the left (or left lying), and lying to the right (or right lying) postures. $T_{i}^{posture}\left(t\right)$ denotes the elapsed time the *i*-th sleeping posture was maintained. In other words, the higher the behavior frequency of the participant to change the sleeping posture, the more likely it is that the NREM sleep stage will begin. These discrimination results were verified by comparing them with the change in heart rate, which was the only biometric information obtained from the participant.
(3)$$g_{snoring}\left(\;t\right)\propto \left(interval_{i}^{snoring}\left(t\right),\;intensity_{i}^{snoring}\left(t\right)\right)\;for\;i\;\in \;N_{snoring}$$

In Equation (3), $N_{snoring}\;$ denotes the number of times the snoring sound exceeded the reference decibel. The sound was recorded for 15 s. $g_{snoring}\left(t\right)\;$ was used to determine the sleep state according to the number of times snoring was detected within a certain period and whether the detection interval was regular or irregular. If the snoring sound is relatively constant and the snoring interval is regular, the sound is used as a criterion for determining the state of NREM sleep.
(4)$$g_{temperature}\left(t\right)\propto \left(temp_{body}\left(t\right),\;temp_{room}\left(t\right)\right)$$

In Equation (4), $temp_{body}\left(t\right)$ and $temp_{room}\left(t\right)$ represent the body and room temperature, respectively. Body or room temperature is a major factor that directly affects sleep status. However, herein, the sleep state of the participant was measured in spring and autumn, seasons that do not require air conditioning, to maintain the monitoring conditions for the non-constraint and unconscious states.

To verify the accuracy of the results of the sleep pattern monitoring proposed here, they were compared with the heart rate of the participant. The ${\mathrm{CO}}_{2}$ concentration increased with the heart rate; however, but it was not used as a direct criterion for determining the sleep state because, although a somewhat large change in ${\mathrm{CO}}_{2}$ concentration occurs around the time of waking up, it is not as clear to recognize it as a meaningful change. Instead, the change in the ${\mathrm{CO}}_{2}$ concentration was used as an indirect criterion to investigate the relationship between the number of tosses and turns and snoring.

### 4.1. Classification of Sleeping Posture Using the Smart Pillow

The distribution of pressure due to the placement of the participant’s head on the smart pillow was analyzed and used to determine the sleeping posture. More than 1 million pressure samples collected from September 2020 to January 2021 constituted the datasets. The sleeping posture was determined by designing a supervised machine learning model (SVM). The pressure data from the smart pillow were collected every second. Typically, the sleeping posture was determined using accumulated pressure data on a minute-by-minute basis. The cycle of discriminating sleep posture was variable, and when a rapid change in pressure was detected, it could be shortened to a minimum interval of 10 s. In the same sleeping posture, light tossing and turning, such as slightly moving only the head, were recognized as one. The threshold $th_{toss-and-turn}$ for judging light tossing and turning is defined in Equation (5).
(5)$$th_{toss-and-turn}=\sum_{i=1}^{8}\left|p_{i}\left(t\right)-p_{i}\left(t-1\right)\right|$$

$p_{i}\left(t\right)$ denotes the pressure value of the *i*-th FSR sensor at time *t.* While determining the sleep state, the duration of tossing and turning is a more important factor than their frequency. If there was no change in the pressure for 5 s, it was considered that tossing and turning had stopped.

The accuracy of posture discrimination using SVM was 94–97% with adjustments to the hyperparameters. Additional verification was conducted by comparing the captured image with the discrimination result. Although the discrimination accuracy changed according to the $th_{toss-and-turn}$ value, we focused on accurately discriminating the duration of tossing and turning, as mentioned above, rather than improving the discrimination accuracy.

Figure 10 shows the average pressure intensity of each sensor in the sleeping posture as determined by the machine learning model. In Figure 10, the number on the *x*-axis represents the eight FSR sensors embedded in the smart pillow. That is, 0 and 7 are the leftmost and rightmost FSR sensors, respectively. In Figure 11, the upper graph depicts the pressure distribution of each sensor, and the lower graph displays the pressure distribution for the fourth and fifth FRS sensors with respect to each sleeping posture.

### 4.2. Relationship between Tossing and Turning and Sleep State

Figure 12 shows the measurement results for 2 h and 49 min from 05:03:46 a.m. to 07:52:45 a.m. on Friday, 22 October 2021, which was when the participant just woke up. Considering the temperature in autumn, the room temperature in the early morning was maintained at 22 °C–23 °C. All temperature units in this paper are in Celsius. In Figure 12, the contactless body temperature suddenly dropped to 33.5 °C–34.0 °C because the participant changed his posture. This is because the contactless temperature sensor measures the temperature of the participant’s head; thus, a change in the posture leads to inaccurate measurements. These measurements show that the sleeping posture of the participant changed twice. Significant changes in the ${\mathrm{CO}}_{2}$ concentration and heart rate were observed at the time just waking up from sleep. During the NREM sleep state, the ${\mathrm{CO}}_{2}$ concentration changed; however, it was judged as not significant.

Figure 13 shows the snapshots of the participant when he was tossing and turning from left to right. In the analysis of the tossing and turning pattern based on the cumulative frequency, a total of three tosses and turns were counted, as shown in Figure 14. The frequency of tossing and turning coincides with the curve of the contactless body temperature, but sleeping behavior can be identified in more detail. In other words, the heart rate barely changes with slow tossing and turning but increases with quick movements. Frequent tossing and turning is considered an REM sleep state rather than an NREM sleep state. For the participant, the frequency of tossing and turning just before waking up rapidly changes; thus, it can be predicted that the participant was in the waking stage.

### 4.3. Relationship between Snoring and Sleep State

Figure 15 is a time series graph showing the recording of a sound from the moment the sound sensor detected that it exceeded the threshold (30 dB). The detected sound was automatically recorded for 15 s. After hearing the recorded sound, it was confirmed that the participant was snoring. The left graph of Figure 16 (01:57:50 a.m. to 01:58:05 a.m.) shows that a total of three snoring events were detected. A delay time of ~2 s was required until the recording restarts. Following this delay, a new recording began again. The right graph of Figure 15 (02:00:50 a.m. to 02:01:05 a.m.) shows that a total of four snoring events were detected. In the left graph, the snoring detection intervals were somewhat irregular; however, in the right graph, snoring was detected at regular intervals of ~3 s. That is, following a short period after snoring began, the participant snored in a relatively stable state.

Further, we examined the relationship between snoring and sleep stages. Figure 16 shows the results of comparing the heart rate measured during snoring with the recorded snoring sound. Heart rate during snoring was lower than at other times. Although snoring-related data can be sufficiently used to analyze sleep apnea, snoring frequency or sound intensity is not directly related to sleep status. However, as shown in Figure 17, a slight change was observed in the ${\mathrm{CO}}_{2}$ concentration during snoring. Thus, the ${\mathrm{CO}}_{2}$ concentration emitted from the body increases during heavy snoring events. Compared with the overall changes in the ${\mathrm{CO}}_{2}$ concentration, the effect of snoring engendered only a minute change.

The user interface screen of the SPMS was configured as shown in Figure 18 to ensure that the observer can monitor the sleep state of the participant. The observer can check ambient environmental data, including light, humidity, fine dust, ${\mathrm{CO}}_{2}$ concentration, room and body temperature, heart rate, and the captured images of the participant’s sleeping posture. A sleep lamp was installed next to the bed to measure light. In addition, the real-time pressure of the smart pillow was demonstrated to evaluate the accuracy of determining the sleeping posture.

### 4.4. Prediction of the Two Sleep Stages: REM and NREM Sleep

This study was conducted to collect unbiased data over a long period of a single participant rather than over a short period with multiple participants. Thus, one of the authors participated in the study, and the experimental data were collected over 2 years. In addition, it was conducted in our laboratory to maintain the experimental environments.

As mentioned earlier, instead of classifying normal sleep into several stages, we predicted only REM and NREM sleep stages using the information collected in the unconstrained and unconscious state. In addition, to confirm the feasibility of the predicted sleep stage, it was compared with the heart rate. When meaningful features were extracted from sleep information, the graphic symbols below were used for visibility.


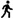
 Tossing and turning.


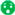
 Snoring in the NREM sleep stage.


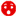
 Snoring in the REM sleep stage.


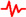
 Bradycardia (slow pulse rate of <60 beats/min) or tachycardia (frequent pulse rate of >100 beats/min).

Sleep-related information in the sections predicted by the NREM and REM sleep stages is shown in Figure 19 and Figure 20, respectively.

Although snoring occurs during both NREM and REM sleep stages, as shown in Figure 19 and Figure 20, it occurs relatively regularly during the NREM sleep stage and irregularly during the REM sleep stage. In addition, the heart rate exhibits minimal changes during snoring and slight tossing and turning, such as head movement. However, if there is even slight arm movement while tossing and turning, the heart rate gradually increases.

In Figure 19, the heart rate of the participant remained low during the NREM sleep stage (maintained for approximately an hour), corresponding to NREM 1 and 2 stages. During this stage, snoring was frequent, 96 times at 3-s intervals. During the NREM sleep stage, a low heart rate was generally maintained.

During the REM sleep stage (Figure 20), bradycardia occurred twice (from 05:28:18 a.m. to 05:34:05 a.m. and from 05:38:47 a.m. to 05:39:34 a.m.). In addition, tachycardia occurred once between 05:39:39 a.m. and 05:40:40 a.m. Bradycardia denotes a very slow heart rate, usually <60 beats/min. An adult’s heart rate usually ranges from 60 to 80 beats/min, with an average of 70 beats. However, even healthy people often exhibit a heart rate of <60 beats/min. Tachycardia is the opposite of bradycardia, i.e., the heart rate is >100 beats/min, which is faster than normal. In Figure 20, the sudden change in the heart rate was very short and signified the waking up of the participant while sleep talking. During the REM sleep stage, which corresponds to REM 1 and 2, the heart rate was relatively high and snoring was irregular.

Figure 21a summarizes the sleep information of the participant obtained while he was asleep. Figure 21b demonstrates the sleeping posture, posture maintenance duration, number of tosses and turns, number of times snoring was detected, heart rate, ${\mathrm{CO}}_{2}$ concentration, room temperature, and humidity for each predicted sleep stage.

The accuracy of the discrimination of the sleep states was compared with that obtained via a mobile phone application; however, providing a specific number was not possible because the latter could not be accurately compared with the clinical test results. Instead, we performed statistical analysis. The area under the curve (AUC) used in the receiver operating characteristic curve is shown in Figure 22. The prediction accuracy was ~90%.

The measurement results of the SPMS were compared with those of the application (named Sleep Cycle) on the smartphone. The operating principle of the mobile application involves predicting sleep via sound and snoring detection. Although the sleeping time measured by this application is accurate, it does not provide a detailed ratio of the REM to NREM stages. Furthermore, as it does not measure heart rate, it cannot differentiate between the NREM and REM stages.

Figure 23 shows the comparison of the measurement results between the mobile application and SPMS to establish the intuitiveness of the SPMS, which involves an unconstrained and unconscious approach. As explained earlier, this application cannot distinguish REM from NREM stages. However, the sleep section identified as REM and NREM sleep overlaps with those predicted by the SPMS. Thus, it is judged to be somewhat shorter than the section predicted as the NREM sleep stage via the SPMS.

### 4.5. Additional Experiments of Two Participants

Additional experiments were conducted for two participants. The first participant, a 56-year-old woman, slept for 5 h and 36 min from 03:25:24 a.m. to 09:01:42 a.m., as shown in Figure 24. She did not snore and changed her posture thrice during sleep. Furthermore, it was determined that she fell into an NREM sleep state immediately after resting following a change in her sleeping position.

Another participant, a 27-year-old man, slept for 6 h and 40 min from 02:48:30 a.m. to 9:28:57 a.m., as shown in Figure 25. In the sleep postures other than the supine posture, he was identified to be in the REM sleep state, during which his heart rate remained very high. Upon inquiry, the participant reported that he had high blood pressure. However, when he was sleeping in the supine posture, his heart rate decreased and he fell into an NREM sleep state. In addition, he snored several times, and similar to the other subject, when the snoring interval became constant, he was judged to have fallen into an NREM sleep state.

For actual verification, the results of the accuracy analysis by applying various machine learning methods using acquired data are summarized in Table 2. The data acquired from the three subjects were compared by sampling data from 1 h to 1 h and 30 min during sleep. Among the machine learning models, ensemble modeling was determined to be the most effective for determining sleep states. Thus, a decision tree model and a random forest model were used. The third subject who woke up in the middle was judged to have relatively low sleep quality, whereas the second subject was judged to have high sleep quality.

## 5. Conclusions

We implemented a prototype system that can monitor the sleep state of individuals in an unconstrained and unconscious state. The sleep state was determined using only the surrounding environmental data and the physical status data of the participants, including snoring, tossing and turning, and body temperature.

All data were acquired in a noncontact manner. Body temperature was measured using a contactless temperature sensor attached to the ceiling of the bedroom. Snoring and ambient noise were measured by configuring the hardware connected to the sound sensor and a microphone installed next to the bed. To determine the sleeping posture, a smart pillow incorporated with eight pressure sensors (FSR-406) in a straight line was utilized. The accuracy of posture discrimination using the smart pillow was 94–97%.

Based on the information collected in a contactless manner, the sleep states were classified into only two stages: NREM and REM sleep. The experimental results showed that the frequency of tossing and turning correlated with the concentration of ${\mathrm{CO}}_{2}\;$. In addition, snoring occurred at regular intervals after a short period elapsed since its initiation. Moreover, snoring exhibited a low correlation to be used as a factor to evaluate sleep status. Snoring was also compared with the measurement results of the sleep analysis smartphone application.

To verify the accuracy of the discrimination results, they were compared with the heart rate of the participant, which was the only biometric information obtained. In addition, for a comparative experiment, the participant wore a smartwatch, and the sleep state was measured using a sleep analysis application in a smartphone wirelessly connected to the smartwatch. The results were then compared with the analysis results of the proposed system.

The results of determining the sleep state in the unconstrained and unconscious state confirmed that the discrimination accuracy of the system was very high. This study provides a foundation for monitoring sleep status while sleeping naturally in a normal living space. There was only one participant, and the experimental data were accumulated for >1 year. When this approach is applied to different individuals, it is expected that various sleep patterns can be derived and a diversity of individual sleep patterns can be detected.

Some limitations associated with this study are as follows: The smartwatch used was Samsung’s Galaxy Watch 4. It has eight built-in functions, including electrocardiogram, blood pressure, and saturation of partial pressure oxygen. According to the smartwatch evaluation of the Korea Consumer Agency in August 2022, it was the product with the most healthcare functions among similar products. Herein, comparing the results of the SPMS with a smartphone application was a means for validation. In other words, it was used to compare whether the analysis results for overall sleep patterns were consistent.

In the current monitoring environment, only one participant can lie on the bed in which the experimental facilities have been installed. We did not consider any disturbances to the measurement results when a pet slept next to the subject.

Furthermore, placing one’s head on a pillow does not determine that the individual is sleeping. The smart pillow was used not to measure sleep time but to determine the frequency of tossing and turning during sleep, and hence, we assumed that individuals lie down with their head on the pillow to sleep. Thus, there may be some errors in the measurement of sleep time.

## Figures and Tables

**Figure 1 sensors-22-09296-f001:**
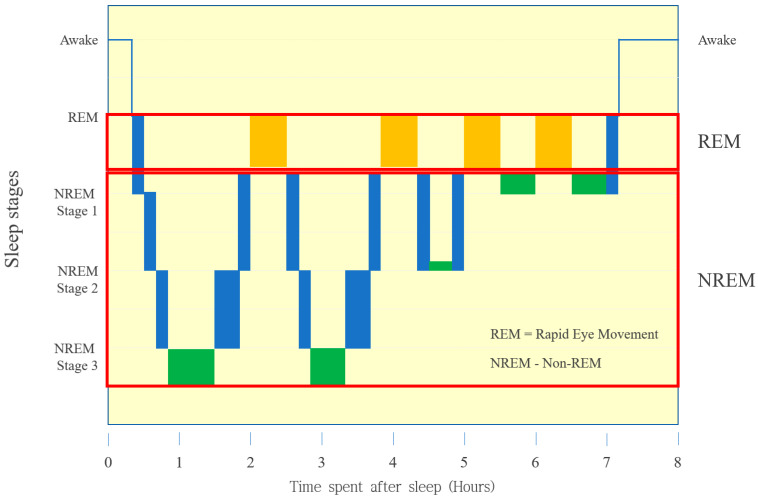
Sleep stages.

**Figure 2 sensors-22-09296-f002:**
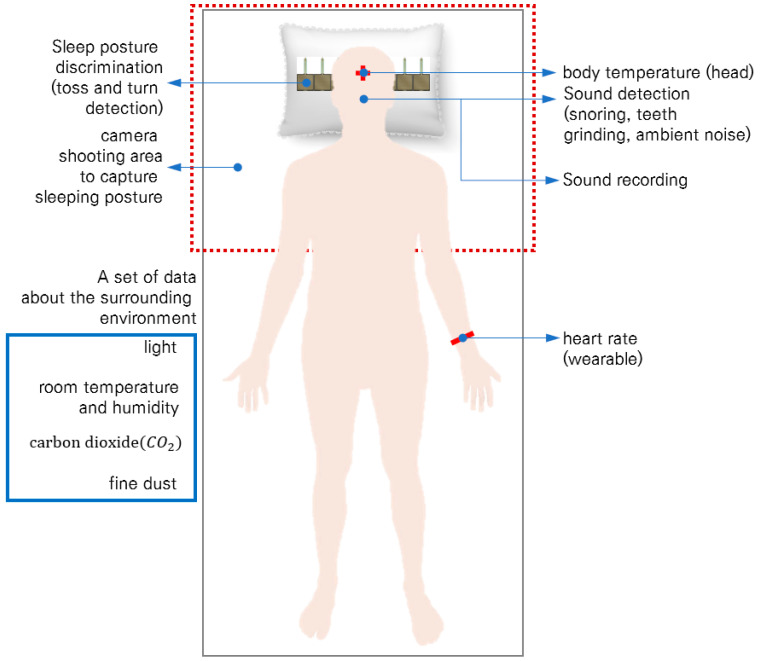
Sleep-related information for sleep pattern analysis.

**Figure 3 sensors-22-09296-f003:**
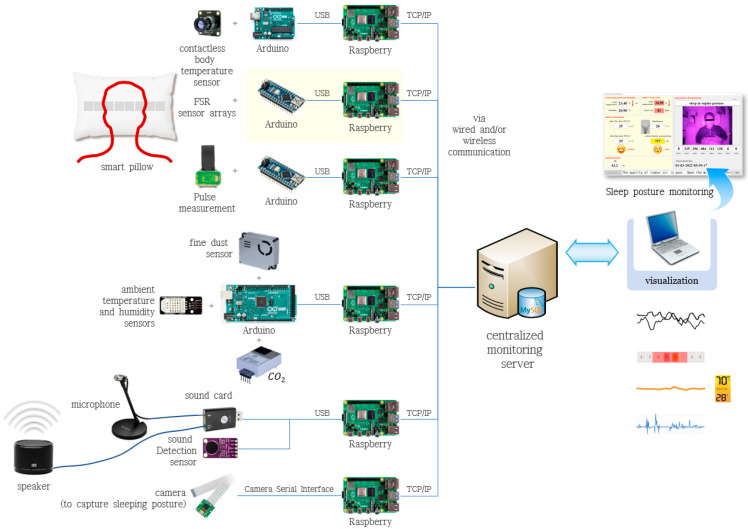
Overall architecture of the sleep pattern monitoring system using nonwearable sensors.

**Figure 4 sensors-22-09296-f004:**
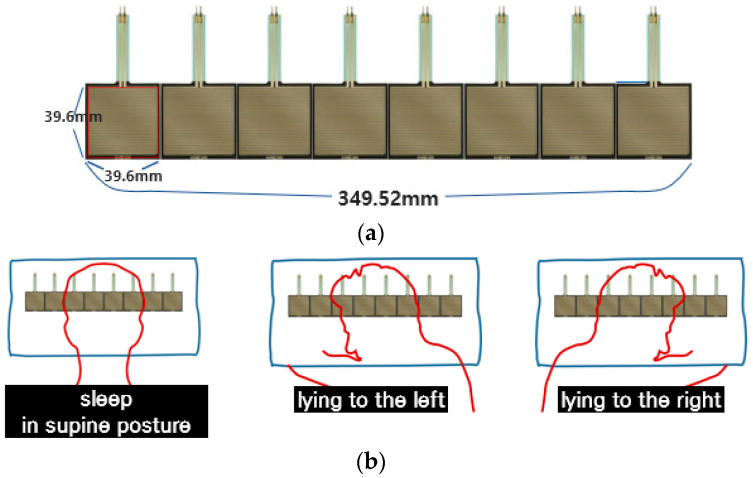
Smart pillow and the three sleeping postures it can distinguish. (**a**) Horizontal arrangement of eight FSR sensors; (**b**) sleeping postures that can be discriminated using the smart pillow.

**Figure 5 sensors-22-09296-f005:**
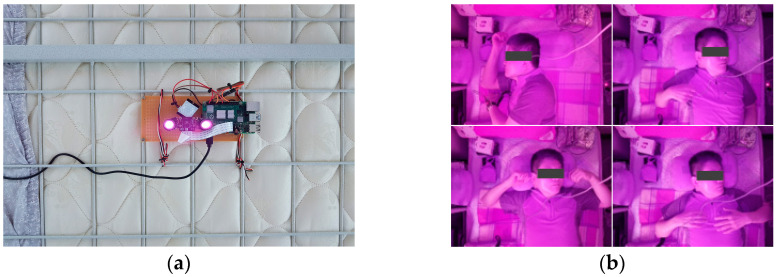
Sampling of sleeping posture images using an infrared camera. (**a**) Infrared camera module; (**b**) sleep posture images.

**Figure 6 sensors-22-09296-f006:**
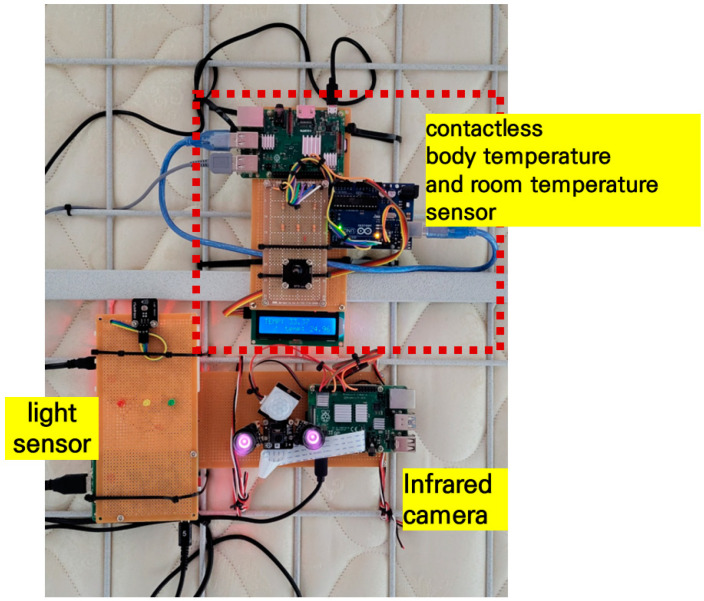
Set of three sensor modules attached to the ceiling.

**Figure 7 sensors-22-09296-f007:**
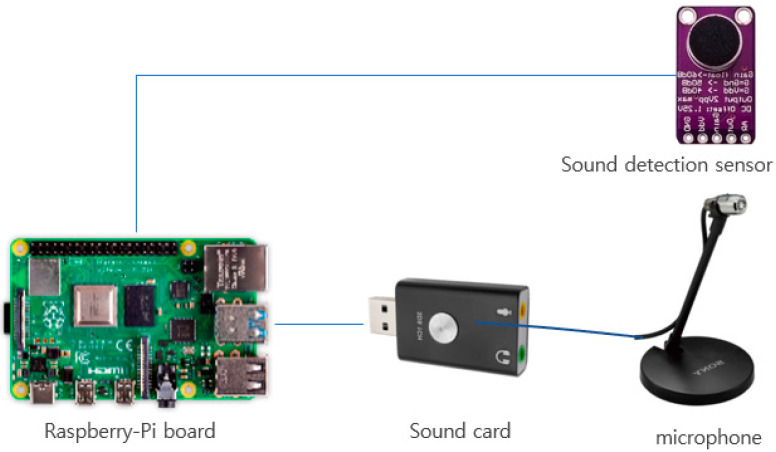
Hardware configuration for sound detection and recording.

**Figure 8 sensors-22-09296-f008:**
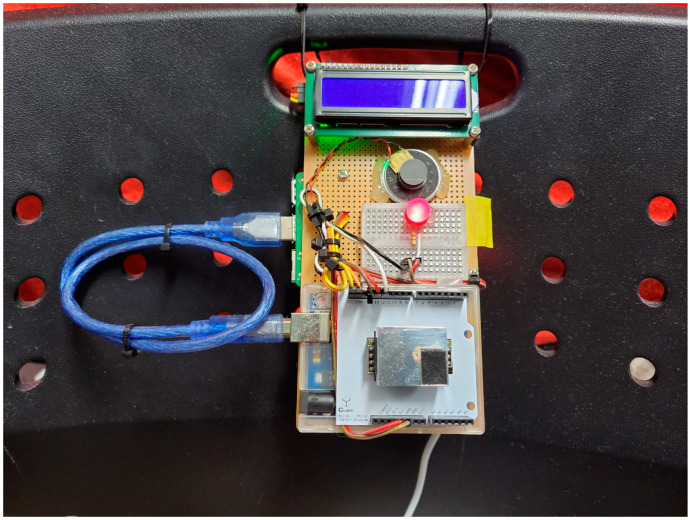
${\mathrm{CO}}_{2}$ concentration measurement module.

**Figure 9 sensors-22-09296-f009:**
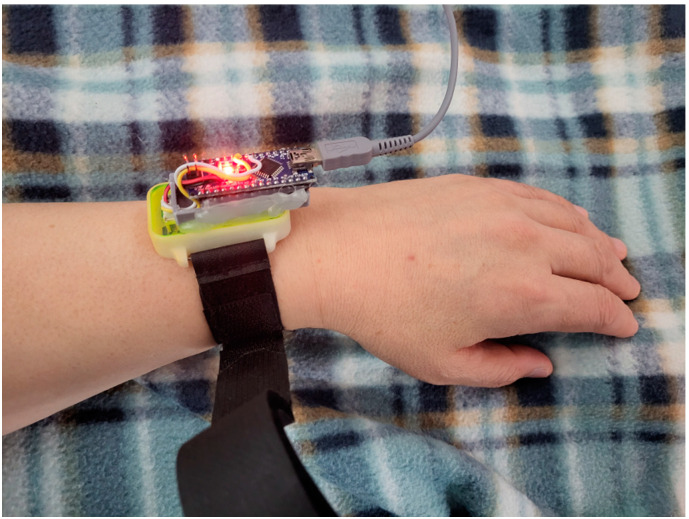
Band-type heart rate-measuring device.

**Figure 10 sensors-22-09296-f010:**
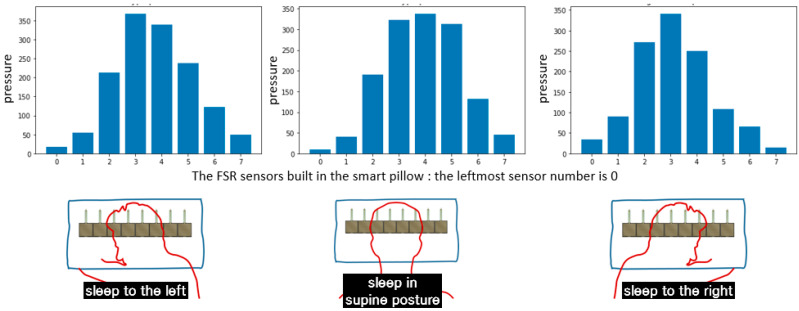
Average pressure intensities of the FSR sensors with respect to the sleeping posture.

**Figure 11 sensors-22-09296-f011:**
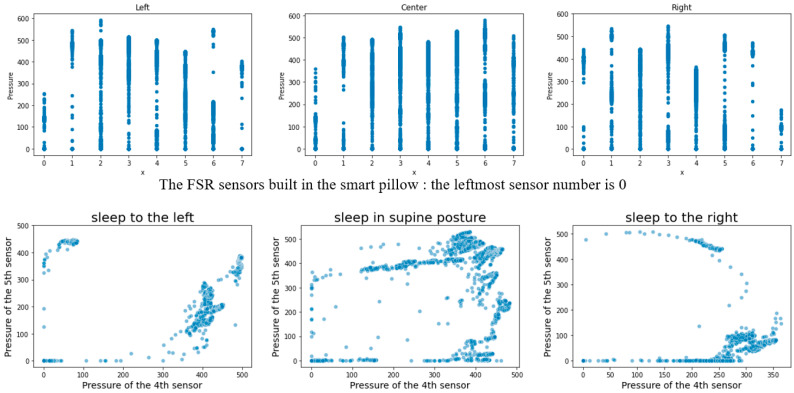
Various pressure distributions for each sleeping posture, each FSR sensor (**above**), and the fourth and the fifth sensors (**below**).

**Figure 12 sensors-22-09296-f012:**
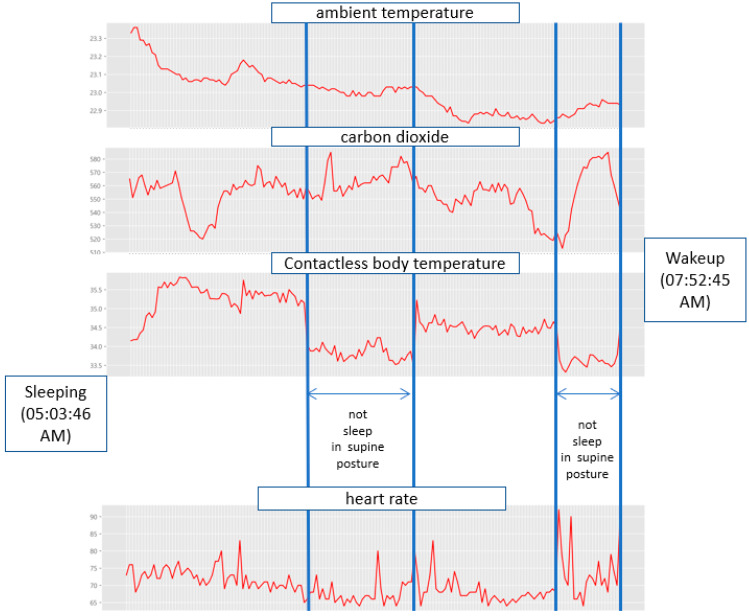
Sleep pattern measurement results (measurement period: 05:03:46 a.m. to 07:52:45 a.m. 22 October 2021).

**Figure 13 sensors-22-09296-f013:**
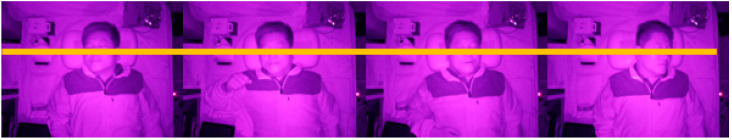
Snapshots of the left-to-right tossing.

**Figure 14 sensors-22-09296-f014:**
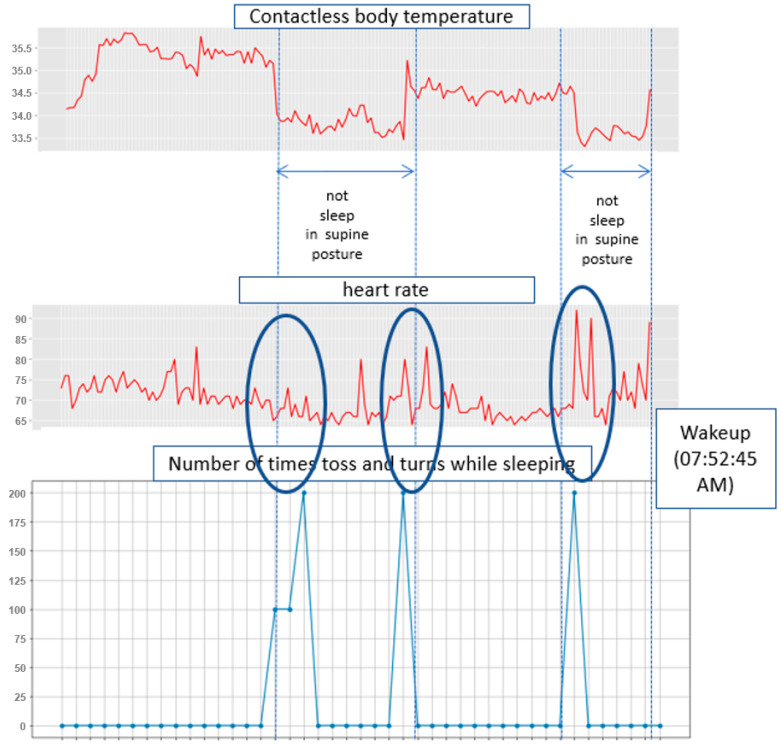
Changes in heart rate according to the frequency of tossing and turning.

**Figure 15 sensors-22-09296-f015:**
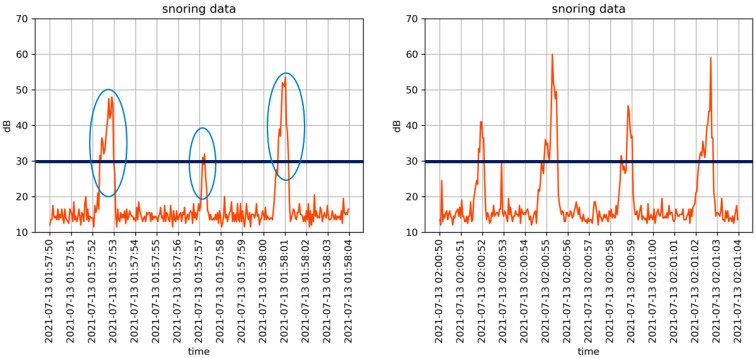
Sound detection of snoring interval (15 s): 01:57:50 a.m. to 01:58:05 a.m. (**left**) and 02:00:50 a.m. to 02:01:05 a.m. (**right**).

**Figure 16 sensors-22-09296-f016:**
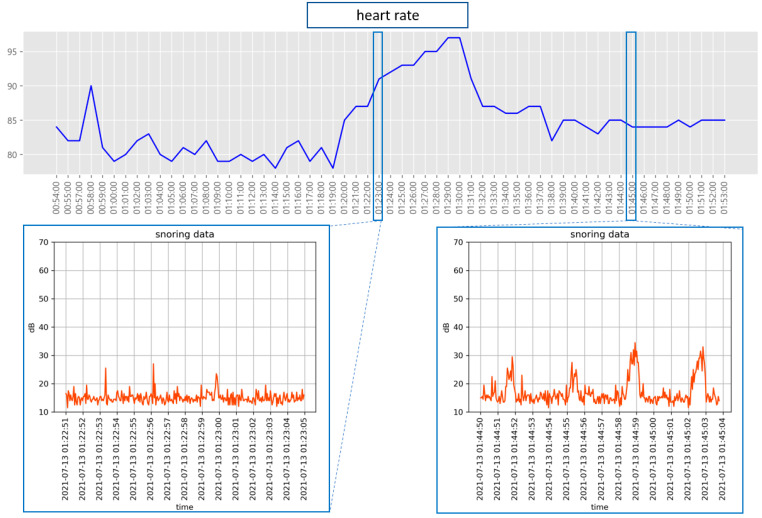
Comparison of snoring sound and heart rate.

**Figure 17 sensors-22-09296-f017:**
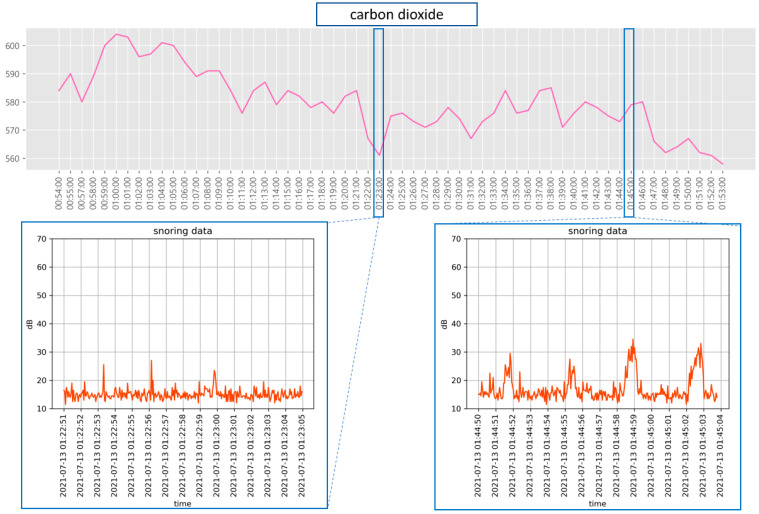
Comparison of snoring sound and carbon dioxide concentration.

**Figure 18 sensors-22-09296-f018:**
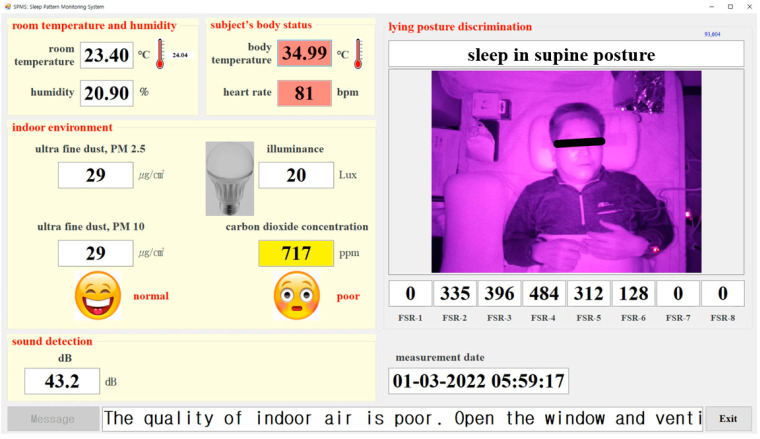
User interface screen of the sleep pattern monitoring system.

**Figure 19 sensors-22-09296-f019:**
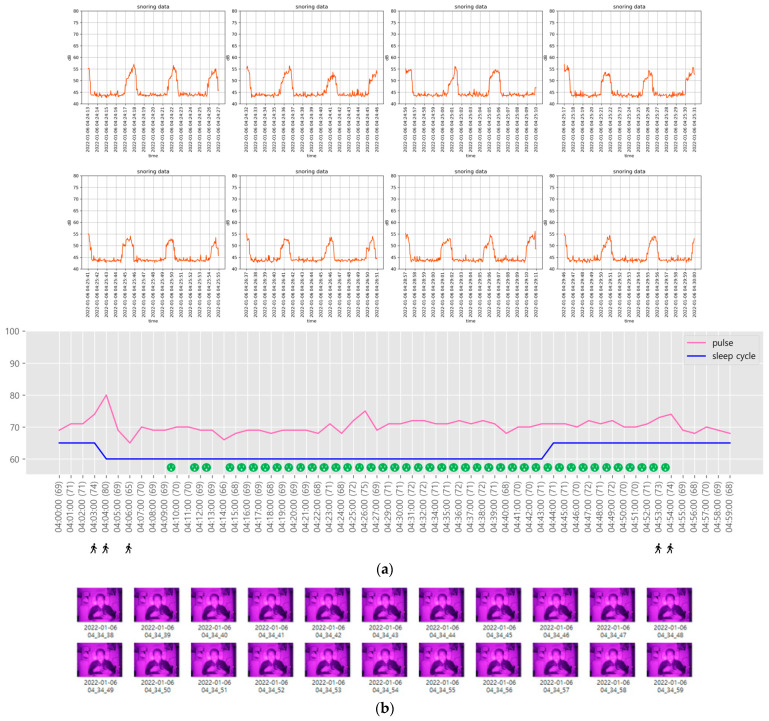
Sleep-related information in the section predicted by the NREM sleep stage. (**a**) Sleep-related information during the predicted NREM sleep stage; (**b**) 22 snapshots of sleep postures captured from 04:34:38 to 04:34:59.

**Figure 20 sensors-22-09296-f020:**
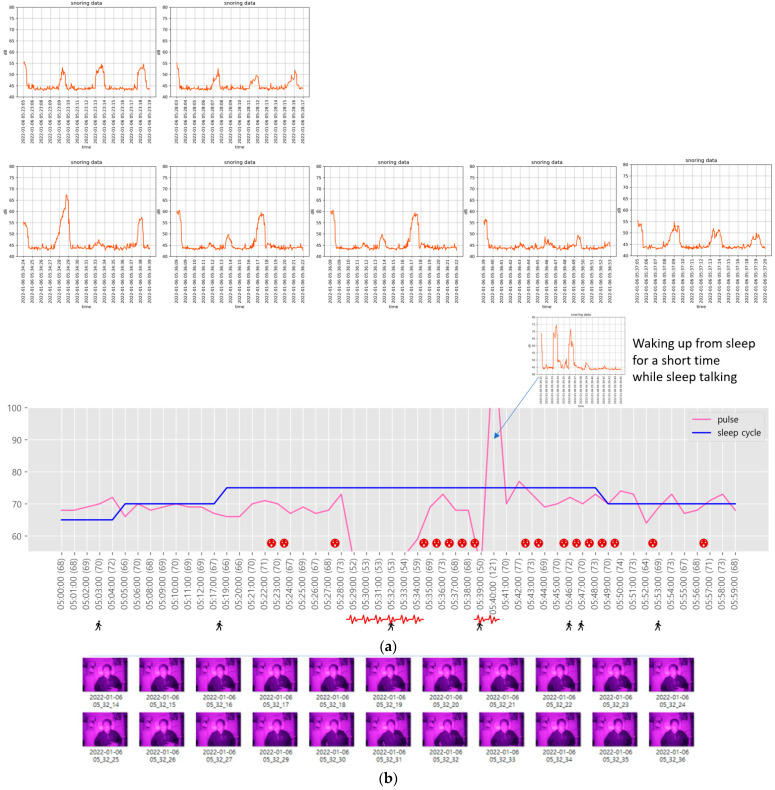
Sleep-related information in the section predicted by the REM sleep stage. (**a**) Sleep-related information during the predicted REM sleep stage; (**b**) 22 snapshots of sleep postures captured from 05:32:14 to 05:32:36.

**Figure 21 sensors-22-09296-f021:**
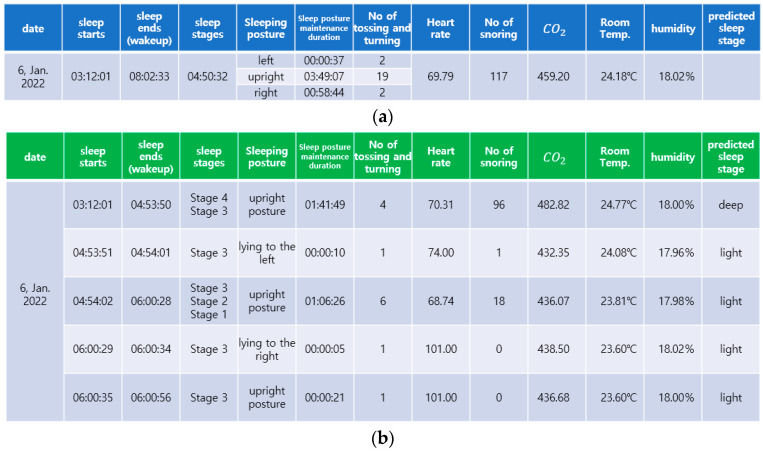
Summary of sleep information obtained during sleep. (**a**) Summary of the participant’s sleep information; (**b**) Summary of sleep information for the two sleep stages.

**Figure 22 sensors-22-09296-f022:**
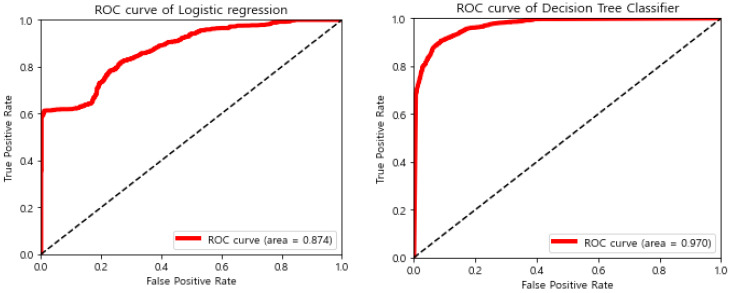
ROC of the logistic regression and decision tree classifiers.

**Figure 23 sensors-22-09296-f023:**
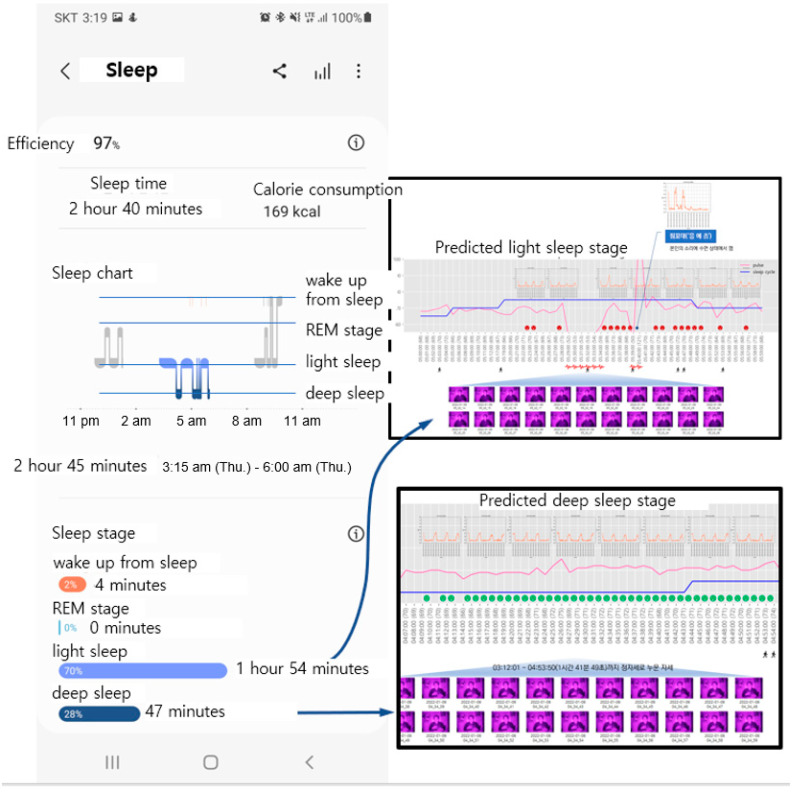
Comparison of the measurement results of the application and SPMS.

**Figure 24 sensors-22-09296-f024:**
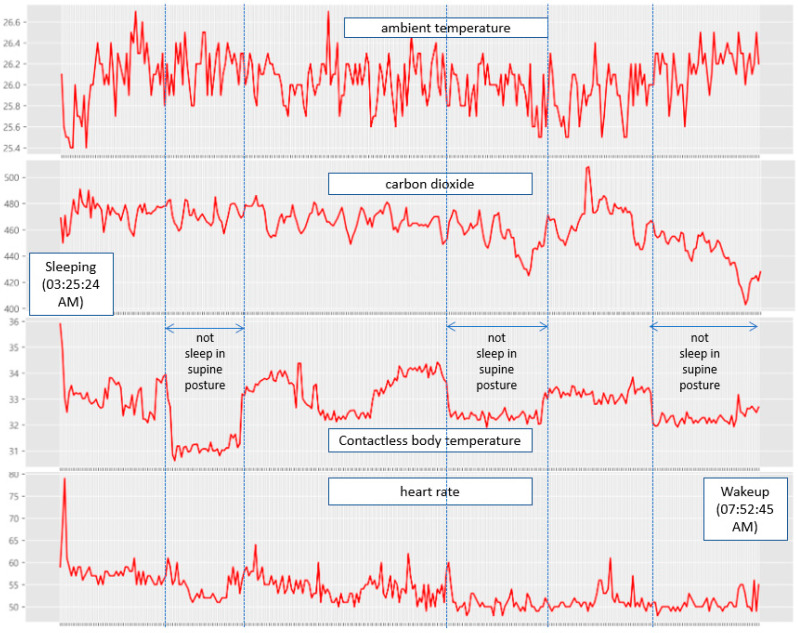
Sleep pattern measurement results of the 56-year-old woman.

**Figure 25 sensors-22-09296-f025:**
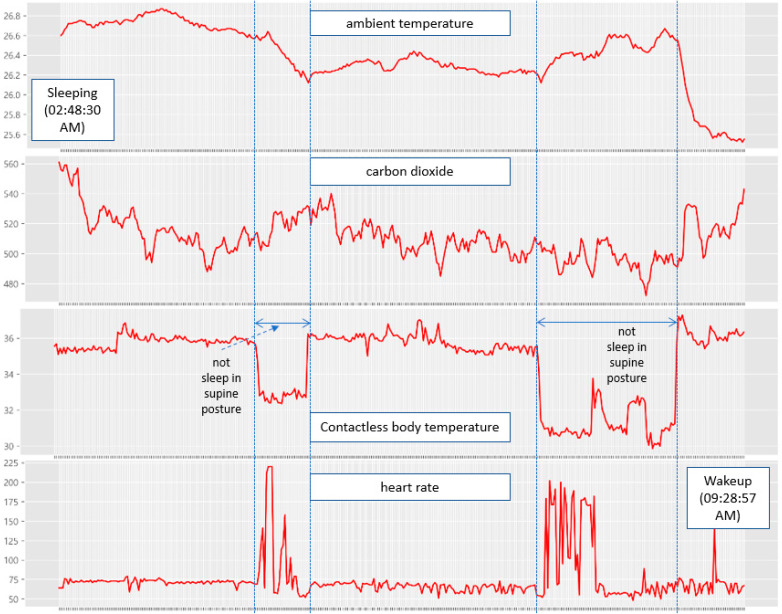
Sleep pattern measurement results of the 27-year-old man.

**Table 1 sensors-22-09296-t001:** Functions and specifications of the sensors used in the monitoring system.

Sensor Type	Functions	Specification
Smart pillow	Total sleep time Sleep posture Discrimination frequency of tossing and turning	FSR-406 pressure sensors range: 100 g–10 kg
Contactless temperature sensor	Body temperature Room temperature	DHT22 (room temperature) Range: −40 °C to 80 °C (±0.2 °C) DTS-L300-V2 (body temperature) Range: −30 °C to 300 °C (±2%) Measurement distance: within 1 m
Heart rate measurement	Heart rate (biological data)	Wearable sensor Serial wire debug interface
Sound	Snoring Teeth grinding Ambient noise	Sound detection (≥30 dB) recoding time: 15 s (maximum)
CO_2_	Status classification with respect to CO_2_ concentration	Classification criteria Good: CO_2_ < 400 ppm Average: 400 ≤ CO_2_ < 700 ppm Poor: 700 ≤ CO_2_ < 1000 ppm Very poor: CO_2_ ≥ 1000 ppm
Light sensor	Brightness	Photosensitive resistor Sleeping time: 7–30 LUX (recommended) daytime: 600–850 LUX evening time: 480–560 LUX
Infrared camera	Sleeping posture (image capture)	8-megapixel resolution: 640 × 480

**Table 2 sensors-22-09296-t002:** Statistical analysis of the monitoring results of the three participants.

	Participant#1 (56 Years Old, Male)		Participant#2 (56 Years Old, Female)		Participant#3 (27 Years Old, Male)	
	Decision Tree	Random Forest	Decision Tree	Random Forest	Decision Tree	Random Forest
**accuracy**	0.903	0.906	0.918	0.918	0.831	0.850
**precision**	0.732	0.733	0.676	0.678	0.735	0.725
**recall**	0.635	0.647	0.521	0.525	0.629	0.645
**F1-score**	0.679	0.689	0.588	0.592	0.678	0.683
**AUC**	0.962	0.971	0.949	0.952	0.949	0.952

## Data Availability

All data used in this paper are dependent on the sensors used and the measurement environments. The measurement values of each sensor used in our experiments will be provided upon request by e-mail.

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
