# Peer review of "Sleep Pattern Analysis in Unconstrained and Unconscious State"

_sensors, 2022, doi:10.3390/s22239296_

Round 1
Reviewer 1 Report
In this paper, the authors demonstrate a prototype system that can monitor the sleep state. The study investigated sleep state by using the surrounding environment data and the physical status data. The system contains a contactless temperature sensor attached to the ceiling, a sound sensor installed next to the bed, and a smart pillow with pressure sensors. There is apparent great value in such wearable device. In my opinion, this work will standout if the following comments could be addressed:
· It’s better to polish Figure 2, especially for sound sensors. In Figure 2, the sound detector was on head, but it was placed installed next to the bed.
· Information in Figure 19 and 20 cannot be read. It’s better to change font size and reorganize the pictures.
· The authors stated the discrimination accuracy of sleep state was very high, but no specific accuracy number was given in the manuscript, except the accuracy of posture as 94% to 97%.
Reviewer 2 Report
Review for “Sleep pattern analysis in unconstrained and unconscious state”
The authors have used a group of sensors to measure human and environmental factors to evaluate sleep, with sleep staging categories of light and deep. To evaluate the system, authors have collected data for many nights from a single subject.
Major comments:
The most important shortcoming of this research is the use of a single subject. The experimental design completely ignores between-subjects variability. I recommend performing these experiments on at least a handful of subjects and rewriting the paper.
It seems this system is designed for at-home use but there in no discussion on how data would be collected and interpreted in case there is another person or pets on the same bed at the time of sleep.
Furthermore, to evaluate some of the functionality of this system a cell phone app (Sleep Cycle) was used. Authors fail to discuss how accurately the cell phone app works and why it was appropriate to be used as a reference method/device. Lastly, visual examination of data is not sufficient to compare the two devices. Authors should use acceptable statistical methods to compare epoch-to-epoch sleep analysis/staging between their proposed system and a reference method. The statistical methods should include a confusion matrix in addition to a report of sensitivity, specificity, accuracy, and Cohen’s kappa.
Minor comments:
It is not clear why the authors chose sleep classification into REM and NREM. Please explain.
On line #60, it is stated “whether a subject is in a light sleep state (i.e., REM state)”. In sleep staging both “light sleep” and “deep sleep” are considered NREM. Please clarify if in this manuscript the two considered stages are light & deep or REM & NREM.
In the 2nd paragraph on page 3, authors reference studies using where different modalities for sleep studies were reported. Authors should also include some studies that have used EEG based devices for sleep study applications.
On page 3, define abbreviation DTB-SVM and HRV
On line #99 OSA stands for obstructive sleep apnea. Please make the correction.
In general introduction seems more suitable for a review article. I recommend authors adjust the introduction such that it is more relevant to work at hand including pros and cons of different components selected for the sleep study system utilized in this article.
On line #198, why was snoring recorded for 15s? What is the significance of this time duration?
It seems that light, CO2 concentration, ambient temperature and humidity, and fine dust were not used for the sleep study itself. Why were they included in the paper and discussed? Please explain.
On page 7 what is the basis for CO2 level classification? Please provide a reference.
On line #243 it is stated that pillow can be used to measure “sleep time” but it seems to me that using pillow alone one can only determine the duration of time the user had their head on the pillow independent of them being sleep or awake.
I am not sure if data from a single subject is sufficient. People use pillows differently and have different head dimensions. What is reported in line #233 is the averages values for adults in a specific part of the world. How would the result from a single subject be applicable to all? Furthermore, there are some subgroups for different sleep positions. For example, “yearner”, “log” and “foetus” fall under the category of side sleep posture whereas “start fish” and “soldier” fall under the category of back sleep posture. How would these variations of the same sleeping positions affect detection by the pillow.
Regarding DTS-L300-V2, does the accuracy of temperature measurement be affected by room temperature given this measurement method. Please discuss.
Authors state “Our experimental results show that if the sound level detected by the sound sensor exceeds threshold (i.e., 30 dB), it can be determined as snoring.” How was this 30dB threshold determined. Is this threshold assumed to be the threshold for snoring for this specific subject or all future studies. If the latter, how can the result from one single subject be used to determine that?
On line #333 it is stated “heart rate has long-time correlations”. Is this autocorrelation? Please clarify.
Regarding equation 1:
What if the user is in a state of quiet wakefulness after waking up in the middle of the night. Would the user be considered sleep, based on this equation, if they are awake but not moving about? Similarly, how does this correlation work for snoring? Does lack of snoring mean that the user is awake?
On line #379-80 authors state “If the snoring sound is relatively constant and the snoring interval is regular, it will be used as a criterion for determining the state of deep sleep.” Why is that and why is that treated differently from irregularly spaced snoring sounds. Hoffstein et al. in “Snoring and sleep architecture” states “Snoring frequency and snoring index were similar during all sleep stages in light snorers, but they were higher during slow-wave sleep in heavy snorers. Wakefulness time after sleep onset and sleep efficiency correlated significantly with the snoring index”. Please discuss.
On line #393-5 authors state “Instead, the change in the concentration of ??2 will be used as indirect criterion to investigate the relationship between tossing and turning and snoring.” Why is that? Please explain and provide more information.
On line #437-8 authors state “This is because the contactless temperature sensor measures the temperature of the subject's head, so changing the posture does not accurately measure it.” Please clarify what this means.
In Figure 12 do the blue vertical lines indicate movement?
Authors state “a slight change was observed in the ??2 concentration at the time snoring was detected.” Does this mean that during the whole duration of sleep every single snoring event was associated with an increase in CO2?
Round 2
Reviewer 1 Report
Authors have addressed most of comments and concerns in the response letter and the revised manuscript. But it is still highly recommended to improve the image quality of figure 19 and 20. The words in the small images are not readable. These two figures may need to be reorganized before publication.
Reviewer 2 Report
The authors have satisfactorily responded to most of my comments but there are still a few issues that need to be addressed before this article is suitable for publication.
Authors have performed two additional experiments as suggested by this reviewer, but it does not seem that they have incorporated the result of those two additional experiments in their statistical analysis as the order of introduction of statistical analysis and data from the two additional subjects suggest. Is that correct? If yes, authors must use data from all their subjects in their statistical analysis. For this they must use data from one randomly selected night from the first subject and then data from the other two subjects and used all for statistical analysis. Authors should not use data from multiple nights for the first subject for the statistical analysis as this would introduce bias. Please revise.
Out of curiosity, the authors use the phrase “upright sleeping posture”. Did they mean to say sleeping in supine position or was the subject really sleeping upright?
Regarding my previous comments and based on either changes that are made to the text of manuscript or in the response to reviewer #2:
Regarding comment #2, could you please add a few lines discussing this issue in the discussion under shortcomings/limitations or future considerations subsection?
Regarding comment #3, the authors state: “accuracy of the discrimination of sleep state was compared with the mobile phone app …” as such it is valid to ask how accurate the cell phone app is. Without knowing how accurate the cell phone app is the reader can only deduce how similar the two devices are, cell phone app and the device described in this manuscript, but they will not know how accurate your device is. The standard of practice is to compare a new device to a gold-standard device or a known device with previously reported accuracy, sensitivity, specificity and … otherwise comparing two devices does not inform readers beyond the extent of similarity between the two devices. So please include a few lines regarding the accuracy of the cell phone device.
Regarding comment #9, authors state: “However, if enough individual sleep-related data is accumulated, it is expected that sleep quality can be analyzed with reasonable accuracy even in the absence of biometric information.” Please remove these lines if there is no data to support this claim.
Regarding comment #13, intention to sleep and being sleep are two different things. It will not be accurate to equate head on pillow to being sleep. Please discuss in the limitation/shortcoming section.
Regarding comment #15 and #16, for the sake of transparency, could you please clarify this in the text as well if you have already not done so? Thank you.
